# Emerging Advances in the Management of Delayed Cerebral Ischemia After Aneurysmal Subarachnoid Hemorrhage: A Narrative Review

**DOI:** 10.3390/jcm14103403

**Published:** 2025-05-13

**Authors:** Shinsuke Muraoka, Takashi Izumi, Masahiro Nishihori, Shunsaku Goto, Issei Takeuchi, Ryuta Saito

**Affiliations:** Department of Neurosurgery, Nagoya University Graduate School of Medicine, Nagoya 466-8560, Japan; izumi.takashi.g0@f.mail.nagoya-u.ac.jp (T.I.); nishihori.masahiro.r1@f.mail.nagoya-u.ac.jp (M.N.); goto.shiyunsaku.i2@f.mail.nagoya-u.ac.jp (S.G.); takeuchi.issei.d1@f.mail.nagoya-u.ac.jp (I.T.); saito.ryuta.b1@f.mail.nagoya-u.ac.jp (R.S.)

**Keywords:** subarachnoid hemorrhage, vasospasm, delayed cerebral ischemia, prevention, neurocritical care

## Abstract

**Background/Objectives:** Aneurysmal subarachnoid hemorrhage (aSAH) remains a life-threatening cerebrovascular event with high rates of mortality and long-term morbidity. Among its complications, delayed cerebral ischemia (DCI) is a major contributor to poor clinical outcomes. Although cerebral vasospasm has traditionally been considered the primary mechanism underlying DCI, recent studies have revealed the multifactorial nature of this condition. This review aims to provide a comprehensive overview of the pathophysiology, preventive strategies, and current treatment options for DCI following aSAH. **Methods:** A narrative literature review was conducted using the PubMed database to identify peer-reviewed articles relevant to the prevention and treatment of DCI following aSAH. The search strategy employed the following terms: (“Subarachnoid Hemorrhage” [MeSH]) AND “Delayed Cerebral Ischemia” AND (“Prevention and Control” [Subheading] OR “Secondary Prevention” [MeSH]). This search strategy was designed to capture studies addressing both pharmacological and non-pharmacological preventive measures for DCI. **Results:** A comprehensive PubMed search identified a total of 113 relevant articles. Among these, 40 publications primarily addressed pharmacological interventions, while 22 focused on neuromonitoring techniques. An additional 20 articles explored the pathophysiological mechanisms underlying DCI, and 15 involved preclinical studies utilizing animal models. The remaining 16 articles encompassed diverse topics, including prophylactic endovascular therapies, newly proposed definitions of DCI, treatment algorithm development, functional outcome analyses, and entries in clinical trial registries. Emerging evidence highlights that vasospasm alone does not account for all cases of DCI. Pharmacological approaches such as nimodipine, clazosentan, and fasudil have shown varying degrees of efficacy. Circulatory management and removal of subarachnoid hematoma via CSF drainage or thrombolytics may reduce DCI risk, although their impact on long-term neurological outcomes remains controversial. Endovascular therapy and adjunctive agents such as cilostazol or anticoagulants have demonstrated potential but require further validation through large-scale trials. **Conclusions:** Effective DCI prevention and treatment require a multimodal approach targeting diverse pathological mechanisms beyond vasospasm. Improved risk stratification, early detection, and individualized therapy are essential for advancing the management of patients with aSAH.

## 1. Introduction

Globally, stroke ranks as the second leading cause of death and disability-adjusted life-years [1]. Intracranial aneurysm rupture, causing aneurysmal subarachnoid hemorrhage (aSAH), accounts for 5% of all strokes [2,3]. The early mortality rate post-SAH exceeds 30%, and most survivors endure long-term disabilities or cognitive complications [2,4]. Cerebral vasospasm affects approximately 70% of aSAH patients, typically manifesting between days 7 and 10 post-bleed [5,6,7]. Delayed cerebral ischemia (DCI), a complex ischemic complication of aSAH, occurs in 17–40% of these patients and is associated with poorer clinical outcomes [8,9,10,11]. Historically, cerebral vasospasm that went unrecognized or was inadequately treated was often cited as a primary cause of mortality in autopsies following aSAH [12]. However, the pathogenesis of DCI extends beyond simple vasospasm, encompassing a spectrum of factors such as disruption of the blood–brain barrier, microthrombosis, cortical spreading depolarizations, and the breakdown of cerebral autoregulation [13]. Interventions that focus solely on vasospasm have demonstrated limited success in ameliorating DCI or reducing the death rate among aSAH patients. Therefore, honing risk stratification methods, improving prophylaxis, and early detection and management of DCI are essential in advancing the treatment of aSAH.

Today, various pharmacological strategies aimed at reducing mortality and DCI while enhancing neurological and functional recovery post-aSAH have been reported. In this review, we summarize the etiology and treatment of post-aSAH DCI.

## 2. Materials and Methods

A narrative literature review was conducted using the PubMed database to identify peer-reviewed articles relevant to the prevention and treatment of DCI following aSAH. The search strategy employed the following terms: (“Subarachnoid Hemorrhage” [MeSH]) AND “Delayed Cerebral Ischemia” AND (“Prevention and Control” [Subheading] OR “Secondary Prevention” [MeSH]). This search strategy was designed to capture studies addressing both pharmacological and non-pharmacological preventive measures for DCI.

## 3. Results

The initial PubMed search yielded 113 published articles. Of these, 40 focused on pharmacological interventions, 22 on monitoring modalities, 20 on the pathophysiological mechanisms of DCI, and 15 on animal experiments. The remaining 16 articles addressed various topics, including prophylactic endovascular treatment (EVT), novel definitions of DCI, treatment algorithms, outcome analyses, and clinical trial registries.

Since this narrative review aims to focus on the prevention and treatment of DCI in clinical settings, studies on animal models and monitoring techniques were excluded from detailed analysis. The selected literature is categorized and discussed according to the specific pharmacological agents and interventional strategies investigated.

## 4. The Etiology of DCI

DCI has been linked to various factors, including cerebral vasospasm, microcirculatory disorder, microthrombi, and cortical spreading depolarization. Figure 1 illustrates the factors associated with DCI.

### 4.1. Cerebral Vasospasm and Subarachnoid Hematoma

The volume and distribution of subarachnoid hematomas are considered risk factors for cerebral vasospasm [14]. In animal models, it has been reported that oxyhemoglobin contained in the subarachnoid hematoma can induce vasospasm, while methemoglobin and bilirubin do not significantly contribute to vasospasm [15,16,17]. The mechanism by which oxyhemoglobin induces cerebral vasospasm may involve the release of reactive oxygen species during its oxidation to methemoglobin, decreased production of Prostaglandin I2, and increased production of Prostaglandin E2 [18]. There are no specific interventions, but draining the subarachnoid space to remove the blood breakdown products may be a sensible approach.

### 4.2. Microcirculatory Disorder

There may be discrepancies between imaging findings of cerebral vasospasm and clinical symptoms, suggesting the involvement of microvascular spasm that is not detectable by imaging. Studies using PET-CT to assess cerebral blood flow and cerebral blood volume have shown that, unlike in cases of carotid occlusion, patients with cerebral vasospasm following SAH do not exhibit peripheral vascular dilation due to autoregulation [19]. In SAH models with blood injected into the cerebral ventricles, narrowing of the microvessels within the brain parenchyma has been observed, suggesting microcirculatory disorder [20]. Microcirculatory disorder can occur within 24 h of SAH onset and pose a risk for subsequent cerebral vasospasm [21].

### 4.3. Microthrombi

In autopsies of SAH patients (n = 29), a more significant number of microthrombi were found in those who developed DCI, suggesting microthrombi as one of the pathological states related to DCI [22]. Furthermore, observational studies following blood markers after SAH onset reported that levels of von Willebrand factor rise within 72 h of onset, and platelet-activating factor increases within four days of SAH, continuing to rise until 14 days post-SAH [23,24]. There is a correlation between the volume of subarachnoid hematoma and increased coagulation activity, and several observational studies have reported a potential link between enhanced coagulation function and DCI [23,25,26,27].

### 4.4. Cortical Spreading Depolarization (CSD)

CSD involves a sustained propagation of neuronal depolarization across the cerebral cortex at speeds of 2–6 mm/min and is associated with cerebral edema. It is also considered a neuroprotective response, inducing contraction and dilation in brain vessels. While CSD is typically not problematic in an undamaged brain, it is observed in 70–80% of severe SAH patients, and recurring CSD episodes in a damaged brain can worsen cerebral ischemia, even in the absence of cerebral vasospasm [28,29,30].

## 5. The Management of DCI

Over the past 60 years, various preventive and therapeutic measures against cerebral vasospasm have been explored, yet more established methods still need to be determined [31,32]. As previously mentioned, various causes contribute to DCI, and approaches to address these have been examined, mainly through randomized controlled trials (RCTs).

## 6. The Prevention of DCI

Table 1 provides an overview of studies published in the past decade.

### 6.1. Circulatory Management

Following SAH, conditions such as central salt-wasting syndrome (CSWS) and syndrome of inappropriate antidiuretic hormone secretion (SIADH) can lead to hyponatremia and reduced circulating blood volume, which increases the risk of DCI [33]. In cases of SIADH, restriction of free water intake is commonly practiced, which, however, might increase the risk of cerebral ischemia [34]. In post-SAH patients, isotonic fluids are typically used to manage circulation, prevent hyponatremia, and reduce circulating blood volume. Previously, the Triple-H therapy, which involved induced hypertension, hypervolemia, and hemodilution, was practiced but failed to demonstrate efficacy [35]. Some observational studies have associated positive fluid balance after SAH with poor outcomes, and the latest guidelines recommend managing infusion to maintain euvolemia [36,37,38]. Complications like neurogenic pulmonary edema and Takotsubo cardiomyopathy can make circulatory management more challenging.

### 6.2. Removal of Subarachnoid Hematoma

The presence of a subarachnoid hematoma is considered a risk factor for cerebral vasospasm, and its removal through drainage is a recognized, albeit invasive, intervention method. Various techniques, such as ventricular drainage, lumbar drainage, and intrathecal administration of thrombolytic agents, have been considered, but most studies are observational. An RCT dividing 210 SAH patients into two groups based on the presence or absence of lumbar drainage showed a decreased DCI. However, there was no significant difference in neurological outcomes after six months [39]. Furthermore, the majority of patients in this study had mild symptoms, and the effectiveness of this intervention for moderate to severe SAH remains unclear.

A meta-analysis of five RCTs considering intrathecal administration of thrombolytic agents in 465 SAH patients reported improvements in cerebral vasospasm, DCI, and neurological outcomes [40]. However, excluding the RCT that combined this treatment with intrathecal nimodipine, no statistically significant differences were observed, indicating that this treatment has yet to be established as standard.

Recently, a randomized clinical trial (EARLYDRAIN) demonstrated that early prophylactic lumbar cerebrospinal fluid drainage after aneurysmal subarachnoid hemorrhage significantly reduced the rate of unfavorable neurological outcomes (modified Rankin Scale score of 3–6) at 6 months [41]. Additionally, lumbar drainage was associated with a lower incidence of secondary infarctions, supporting its role as an effective adjunct to standard care in improving functional outcomes.

### 6.3. Calcium Channel Blockers

Nimodipine, a dihydropyridine calcium channel blocker, is sanctioned by the US Food and Drug Administration for neurological improvement for post-aSAH patients. A Cochrane review assessing treatments involving calcium channel blockers, including their combinations, comprising 16 RCTs with 3661 participants, showed that these drugs improved neurological outcomes [42]. However, stratified analyses revealed that while oral nimodipine exhibited similar beneficial effects, intravenous nimodipine and other calcium channel blockers did not improve outcomes. RCTs reporting a reduction in DCI with oral nimodipine did not show a decrease in cerebral vasospasm [43]; thus, other neuroprotective actions such as microvascular dilation, reduction in platelet aggregation, and decreased calcium-dependent excitotoxicity are considered. Similar findings have been reported for nicardipine, another L-type dihydropyridine calcium channel blocker. High-dose continuous nicardipine administration (0.15 mg/kg/hr) reduced cerebral vasospasm [44], but there was no difference in neurological outcomes after three months, potentially due to hypotension affecting cerebral perfusion. An RCT comparing low-dose (0.075 mg/kg/hr) with high-dose (0.15 mg/kg/hr) nicardipine found no difference in DCI incidence (~31%) or neurological outcomes after three months in either group, and more adverse events led to early termination in the high-dose group [45].

Calcium channel blockers are effective in preventing DCI following aSAH; however, they may induce hypotension, potentially compromising cerebral perfusion pressure (CPP). In such cases, vasopressors or other circulatory agents may be required to maintain adequate CPP. The optimal CPP should be individually assessed and tailored to each patient’s clinical condition.

### 6.4. Clazosentan

Previous studies have suggested that endothelin-1, a potent and persistent endogenous vasoconstrictor, plays a role in the development of cerebral vasospasm [46]. After aSAH onset, the concentration of endothelin increases in the cerebral arteries, thereby increasing the sensitivity to endothelin-1 and intracellular calcium concentration [47].

Clazosentan, a selective endothelin receptor antagonist, inhibits endothelin-mediated cerebral vasospasm [48]. Several trials have evaluated the efficacy and safety of clazosentan in preventing or modulating cerebral vasospasm in patients with aSAH [49,50,51,52,53]. Clazosentan was recently assessed in placebo-controlled, randomized, double-blind studies in adult Japanese patients with aSAH and showed a significant reduction in cerebral vasospasm-related morbidity and all-cause mortality within six weeks post-aSAH [54]. In real-world data analysis, clazosentan significantly reduced the incidence of vasospasm-related morbidity and related the improvement of prognosis in patients with aSAH [55,56].

Although clazosentan has demonstrated clinical utility in Japan, international trials have indicated that, despite its preventive effect on DCI, it is associated with a high incidence of adverse events and has not shown significant benefit in improving clinical outcomes. However, discrepancies in study design—such as differences in dosing regimens and concurrent use with nimodipine—highlight the need for further investigation.

### 6.5. Fasudil Hydrochloride

Fasudil hydrochloride, a Rho kinase inhibitor, exerts vasodilatory effects by antagonizing intracellular calcium ions and myosin light chain and is involved in processes such as vasoconstriction, activation of inflammatory cells, and endothelial cell damage. An RCT comparing fasudil with placebo over 14 days in post-SAH patients showed decreased DCI and improved neurological outcomes after one month [57]. Another RCT comparing oral nimodipine to a control showed no difference in DCI but improved neurological outcomes after one month [58]. A meta-analysis of eight such RCTs reported similar results, although the sample sizes were generally small, indicating a need for larger-scale RCTs to substantiate these findings [59].

### 6.6. Antiplatelet and Anticoagulant Drugs

A 2007 Cochrane review examining the efficacy of antiplatelet drugs in SAH patients (7 RCTs, n = 1385) found no significant differences in DCI, neurological outcomes, or hemorrhagic complications [60]. Furthermore, although several observational studies have reported on the effectiveness of dual antiplatelet therapy (DAPT), results are inconsistent, and there is a suggested risk of increased bleeding [61,62].

Cilostazol, a phosphodiesterase inhibitor, possesses antiplatelet, endothelial protective, and vasodilatory effects. Several RCTs have reported improvements in cerebral vasospasm, DCI, and neurological outcomes with cilostazol, though the sample sizes were small, indicating a need for verification through larger-scale studies [63,64,65].

Regarding anticoagulant therapy, two RCTs comparing subcutaneous low molecular weight heparin injections with placebo reported reductions in DCI and stroke but no difference in neurological outcomes after three months [66,67]. Continuous infusion of heparin has been observed in studies to improve DCI outcomes, stroke outcomes, and cognitive function after three months [68,69,70].

Given that microthrombi contribute to the pathophysiology of DCI, the potential efficacy of antiplatelet therapy warrants investigation, not only with cilostazol but also with other agents.

### 6.7. Statins

Statins are believed to have anti-inflammatory, vasodilatory, and antithrombotic effects. A meta-analysis of six randomized trials on statin therapy for aSAH revealed a decrease in vasospasm without significant improvement in DCI or mortality [71]. One hypothesis is that extended statin dosing may increase the risk of bacteremia within this patient group, neutralizing any vasospasm-mitigating benefits [72]. Research efforts are pivoting towards examining shorter-duration and combination statin therapies [73]. Given that existing dosing regimens have not yielded outcome benefits, routine statin use for aSAH patients is not currently advised [74].

### 6.8. Magnesium

Although preclinical studies posited that magnesium sulfate could augment cerebral blood flow and curtail vasospasm [75], clinical evidence has not demonstrated outcome benefits with its intravenous administration [76]. There is ongoing debate regarding the significance of magnesium concentration within the cerebrospinal fluid compared to peripheral levels, a theory yet to be empirically substantiated. Based on present data, magnesium sulfate is not endorsed for routine use to enhance neurological outcomes post-aSAH [77].

#### The Treatment of DCI

After the onset of DCI, if there are no signs of fluid overload, fluid loading is first performed to see if it improves neurological outcomes (cerebral blood flow responsiveness to infusion) [78]. Once the circulating blood volume is optimized, if there are no concerns like untreated aneurysms, the target blood pressure is set higher, and vasoconstrictors are administered. Reports suggest these management strategies improve neurological outcomes in about 70% of DCI cases [79].

Endovascular treatment is divided into mechanical dilation using a balloon and intra-arterial administration of vasodilatory drugs. It is considered for patients whose medical treatment is ineffective or poses a high risk of complications. An RCT evaluating prophylactic percutaneous transluminal cerebral balloon angioplasty (PTCBA) within 96 h of SAH onset in patients at high risk of DCI due to large subarachnoid hematomas showed no difference in the occurrence of DCI or neurological outcomes, suggesting that preventive PTCBA is not recommended [80]. Additionally, complications related to PTCBA included arterial perforation in four cases, three of which were fatal. No RCTs have been reported regarding endovascular treatments as interventions for DCI. Observational studies have shown them effective in over 90% of cases, but approximately 20% required retreatment due to symptom recurrence [81]. Endovascular treatment should also be conducted earlier after the onset of DCI [82]. Vasodilatory drugs used intra-arterially include verapamil, nicardipine, nimodipine, and fasudil [83].

Delayed Cerebral Ischemia Following Aneurysmal Subarachnoid Hemorrhage: Current Gaps and Future Research Priorities.

DCI remains one of the most challenging and consequential complications following aSAH, contributing significantly to adverse neurological outcomes. Despite extensive research efforts over the past several decades, the underlying pathophysiological mechanisms, diagnostic criteria, and optimal therapeutic strategies for DCI remain incompletely elucidated. In this context, several key research priorities have emerged that aim to bridge existing knowledge gaps and improve patient outcomes.

Standardization of DCI Definition and Diagnostic CriteriaA universally accepted and clinically relevant definition of DCI is urgently needed. Current diagnostic paradigms vary widely across institutions and studies, incorporating clinical deterioration, radiographic evidence of infarction, or a combination thereof. This heterogeneity impedes cross-study comparisons and limits the generalizability of findings. The development and widespread adoption of standardized diagnostic criteria that integrate both clinical and imaging elements is essential. Furthermore, the incorporation of molecular biomarkers—including indicators of inflammation, thrombosis, and cerebral metabolic activity—may improve diagnostic accuracy. The application of machine learning techniques to synthesize multimodal data into predictive algorithms represents a promising strategy for the early detection of DCI.Toward Personalized Therapeutic StrategiesThere is a growing recognition of the need for individualized therapeutic approaches. Inter-patient variability in treatment response is well documented, rendering standardized treatment paradigms suboptimal. Advances in pharmacogenomics, such as cytochrome P450 polymorphism analysis, may enable personalized dosing regimens, particularly for agents like nimodipine and clazosentan. Furthermore, emerging analytic methodologies, including causal inference techniques and machine learning tools such as causal forests and SHAP (SHapley Additive exPlanations) analysis, offer powerful means to characterize treatment effect heterogeneity and support precision medicine frameworks. Stratifying patients according to predicted therapeutic responsiveness is an essential step toward optimizing individualized care.Identification and Evaluation of Novel Therapeutic TargetsThe discovery and rigorous assessment of novel pharmacological agents remain central to future research. While nimodipine is the only drug with consistent evidence for outcome improvement, its effect size remains modest. Investigational agents targeting inflammatory cascades, platelet aggregation, and endothelial dysfunction merit further exploration. Preliminary data suggest that cilostazol, tirofiban [84,85], nadroparin [86], and dapsone [87] may confer benefits in reducing DCI incidence or enhancing functional recovery. Additionally, novel targets such as soluble epoxide hydrolase inhibitors (e.g., GSK2256294) [88] hold promise, especially in trials employing biomarker-guided enrichment strategies.Advancement of Non-Invasive Neuromonitoring TechnologiesThe development and integration of real-time, non-invasive neuromonitoring modalities may transform DCI detection and management. Current monitoring techniques are often invasive, sporadic, or operator-dependent. Modalities including transcranial Doppler ultrasonography, near-infrared spectroscopy, electroencephalography, and cerebral microdialysis have demonstrated clinical utility but require further standardization and technological integration. The future lies in the creation of bedside, AI-assisted platforms that consolidate physiological and biochemical data to enable dynamic risk stratification and timely therapeutic interventions.Emphasis on Preventive Multimodal StrategiesGiven the multifactorial nature of DCI pathogenesis—including circulatory dysregulation, neuroinflammation, microthrombosis, and impaired autoregulation—prevention must involve multifaceted approaches. Optimization of systemic hemodynamics, early initiation of neuroprotective agents, and prophylactic endovascular techniques should be actively pursued. Intervening during the critical period of vulnerability, typically between days 4 and 10 post-hemorrhage, is of strategic importance.Focus on Long-Term, Patient-Centered OutcomesFuture research should place greater emphasis on long-term functional and quality-of-life outcomes. Traditional endpoints such as the 90-day modified Rankin Scale may not fully capture the breadth of patient recovery. Incorporating measures such as return to work, health-related quality of life (e.g., EQ-5D), and patient-reported outcomes (PROMs) will yield a more comprehensive assessment of long-term recovery. Longitudinal studies linking DCI with societal reintegration and sustained functional independence are warranted.Promotion of International Collaboration and Data HarmonizationMany existing studies are constrained by limited sample sizes, single-center designs, and methodological heterogeneity. Establishing multicenter prospective registries and conducting individual patient-level meta-analyses are critical for generating high-quality evidence and informing global clinical practice guidelines. International data-sharing initiatives are indispensable to accelerate discovery and facilitate the development of universally applicable, evidence-based management strategies.

## 7. Conclusions

DCI remains a complex and critical complication of aSAH, representing a major determinant of poor neurological and functional outcomes. Despite notable advances in the elucidation of its multifactorial pathophysiology—including cerebral vasospasm, microthrombosis, microvascular dysfunction, and cortical spreading depolarizations—therapeutic progress has been modest, and many interventions have yielded inconclusive results.

Among the available pharmacological treatments, nimodipine remains the only agent consistently associated with improved neurological outcomes. Its clinical efficacy appears to extend beyond the amelioration of vasospasm, likely involving neuroprotective mechanisms. Clazosentan, a selective endothelin receptor antagonist, has demonstrated potential in reducing vasospasm-related morbidity and mortality, particularly in Japanese cohorts. Meanwhile, fasudil and cilostazol have shown encouraging results but require validation through adequately powered, multicenter randomized controlled trials. Other therapeutic agents—including statins, magnesium sulfate, and various anticoagulants—have produced mixed or limited evidence of benefit.

Preventive strategies continue to evolve, with current efforts focusing on systemic circulatory optimization, early hematoma evacuation, and prophylactic endovascular interventions. Nevertheless, substantial inter-individual variability in treatment response underscores the limitations of uniform therapeutic approaches. Precision medicine, supported by emerging causal inference frameworks and machine learning algorithms such as SHAP and causal forests, holds promise for tailoring interventions to individual patient profiles and enhancing therapeutic efficacy.

Moving forward, several research imperatives warrant focused attention. These include the following: (1) the establishment of standardized and universally accepted definitions for DCI; (2) the development of biomarker-driven, personalized treatment strategies; (3) the integration of non-invasive, real-time neuromonitoring technologies into clinical practice; and (4) a paradigm shift toward long-term, patient-centered outcome assessment that extends beyond traditional metrics such as the modified Rankin Scale.

International collaboration will be indispensable in overcoming current limitations associated with single-center studies and heterogeneous methodologies. The formation of prospective multicenter registries and global data-sharing platforms will be critical to the generation of high-quality evidence and the formulation of robust, evidence-based clinical guidelines for the prevention and management of DCI following aSAH.

## Figures and Tables

**Figure 1 jcm-14-03403-f001:**
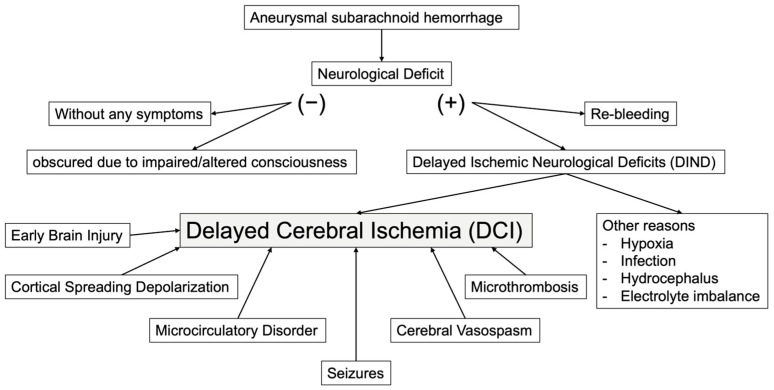
The etiology of delayed cerebral ischemia after aneurysmal subarachnoid hemorrhage.

**Table 1 jcm-14-03403-t001:** The prevention of delayed cerebral ischemia after aneurysmal subarachnoid hemorrhage.

Drugs	Sample Size/Study Type	Primary Endpoints	Effect on DCI Prevention	Improvement in Functional Outcome	Comments
Nimodipine	Intra-arterial administration (n = 10) + pharmacogenomic review	Vasospasm, cerebral oxygenation, GOS	Yes (vasospasm prevention, improves pO₂)	Yes (all patients had favorable GOS)	Gold standard treatment with proven outcome benefit.
Nicardipine Implant	Meta-analysis (n = 284) + NMA (n = 866)	DCI, mRS, angiographic vasospasm	Yes (OR 0.21–0.30 in meta-analysis)	Trend toward improvement (not significant)	Effective adjunct in clipping cases; outcome benefit inconclusive.
Clazosentan	RCT (n = 409) + retrospective (n = 221) + meta-analysis (n = 2778)	DCI incidence, mRS, rescue therapy	Yes (strong; reduces vasospasm and DCI)	Mixed (no effect in RCTs, benefit in retrospective study)	Prevents vasospasm and DCI but increases adverse events; functional benefit limited.
Fasudil	Retrospective study (n = 221, PS-matched n = 27)	DCI, vasospasm, 6-month mRS	Limited (less effective than clazosentan)	Some improvement reported	Traditional Japanese treatment with limited efficacy.
Cilostazol	Meta-analysis including RCTs (n = 543)	Symptomatic vasospasm, infarction, poor outcome	Yes (reduces CV, infarction, DCI)	Yes (OR 0.40 for good outcome)	Low-risk drug with significant effect on both DCI and functional recovery.
Statins	Meta-analysis (n = 1885)	DCI, mRS, mortality	Yes (short-term use effective)	No clear improvement	Suppresses DCI, but no consistent improvement in outcomes.
Magnesium	RCT + retrospective (~n = 250)	DCI, mRS, vasospasm incidence	Uncertain (mixed results)	Some improvement in retrospective reports	Mixed findings; not recommended for routine use yet.
Tirofiban	RCT (n = 30) + retrospective (n = 102)	DCI, vasospasm, bleeding events	Yes (significant reduction in small RCT)	Improvement seen in small RCT	Reduces vasospasm and DCI; bleeding risk must be considered.
Nadroparin	Retrospective cohort (n = 158, high- vs. low-dose)	DCI, mortality, discharge destination	No for DCI, but reduced mortality and better discharge	Yes (lower mortality, better disposition)	High-dose group showed reduced mortality; no increase in adverse events.
Dapsone	Double-blind RCT (n = 48)	DCI, cerebral infarction, mRS (discharge, 3 mo)	Yes (DCI: 26.9% vs. 63.6%, *p* = 0.011)	Yes (favorable mRS and lower infarction rate)	Promising results in all endpoints; needs validation in larger trials.
GSK2256294	Phase Ib RCT (n = 19)	Safety, DCI, biomarkers	Possibly (trends in small trial; needs further study)	Not assessable; further trials needed	sEH inhibitor with good safety profile; further investigation warranted.

CV, cerebral vasospasm; DCI, delayed cerebral ischemia; GOS, Glasgow outcome scale; mRS, modified Rankin Scale; OR, odds ratio; RCT, randomized control trial; sEH, soluble epoxide hydrolase.

## Data Availability

No datasets were generated or analyzed during the current study.

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
