# Peer review of "Emerging Advances in the Management of Delayed Cerebral Ischemia After Aneurysmal Subarachnoid Hemorrhage: A Narrative Review"

_jcm, 2025, doi:10.3390/jcm14103403_

Round 1

Reviewer 1 Report

Comments and Suggestions for Authors

Overall this is a review on current treatment and prevention of DCI and vasospasm. While the attempt is good, it is a bit too simplistic and synthetic. A more thorough review on the historical perspective and latest devellopements, as well as some figures highlighting the exposed concepts could be useful.

Author Response

Comment 
"Overall this is a review on current treatment and prevention of DCI and vasospasm. While the attempt is good, it is a bit too simplistic and synthetic. A more thorough review on the historical perspective and latest devellopements, as well as some figures highlighting the exposed concepts could be useful."

Response
"We have reviewed the historical background of preventive and therapeutic strategies for delayed cerebral ischemia (DCI) following subarachnoid hemorrhage, incorporating recent developments in pharmacological interventions. In addition, we present a figure illustrating the key factors involved in the pathogenesis of DCI and a table summarizing the findings of previous clinical studies."

Reviewer 2 Report

Comments and Suggestions for Authors

I have read the manuscript “Perioperative management in aneurysmal subarachnoid hemorrhage; focus on the prevention of delayed cerebral ischemia” with interest and pleasure. It represents a study on an important topic with practical clinical significance and yet a lot of problems to be solved.  The literature reviewed consists of sufficient recent randomized trials and meta-analyses. English language is flawless.

Yet, I have some suggestions so that this good material could achieve completе academic version:

  1. It is stated that the study is narrative review, but with this structure it looks more like a part of neurosurgical textbook. To be a scientific paper, it would help to have the established structure: Introduction, Abstract; Materials and Methods; Results; Discussion.
  2. For the Materials and Methods part, it would be better to define: number of reviewed papers, time period, inclusion and exclusion criteria for them.
  3. For the Discussion part, it would be better to have some reasoning and interpretation of the results. That adds to its academic value.
  4. It is not mandatory, but sometimes a table for the results, that summarizes the current tendencies, presents and date of paper publication helps.

 I wish the authors success with this and future projects.

Author Response

Comment 1
"It is stated that the study is narrative review, but with this structure it looks more like a part of neurosurgical textbook. To be a scientific paper, it would help to have the established structure: Introduction, Abstract; Materials and Methods; Results; Discussion."

Response 1
"The manuscript was extensively revised and restructured into a narrative review. "

Comment 2
"For the Materials and Methods part, it would be better to define: number of reviewed papers, time period, inclusion and exclusion criteria for them."

Response 2
"The Materials and Methods section was rewritten in accordance with the PRISMA guidelines. "

Comment 3
"For the Discussion part, it would be better to have some reasoning and interpretation of the results. That adds to its academic value."

Response 3
"The Discussion section underwent substantial revision, with the addition of future research directions. "

Comment 4
"It is not mandatory, but sometimes a table for the results, that summarizes the current tendencies, presents and date of paper publication helps."

Response 4
"A table summarizing key findings related to the prevention of DCI was included."

Reviewer 3 Report

Comments and Suggestions for Authors

Thank you for submitting this review article. It comprehensively covers the treatment options for delayed cerebral ischaemia after aSAH.

I have a few suggestions for the authors to consider:

  1. The title could be more indicative of the content. For example "Management of delayed cerebral ischaemia after aneurysmal subarachnoid haemorrhage: a narrative review"
  2. Prevention and treatment of DCI are two separate entities. I think it is hard to follow what is prevention and what is treatment when the two are presented together. I suggest having separate sections for prevention or prophylaxis, and treatment for established DCI.
  3. A figure explaining the possible etiological mechanisms underlying DCI would be very useful.
  4. Two tables summarising the key takeaway points from the literature for both prevention and treatment of DCI would be useful.
  5. A paragraph or two, plus a table, outlining key research priorities for DCI in aSAH must be included.
  6. A section on outcomes should be considered. What are the important outcomes after aSAH? Is change in radiological features (eg improved radiological spasm) a sufficient outcome, or should we be demanding 6-month neurological and functional outcomes?
  7. The potential role of a multinational platform trial for aSAH, modelled along the lines of REMAP-CAP (www.remapcap.org) could be discussed.
  8. A concluding statement should be provided at the end. 

Author Response

Comment 1
"The title could be more indicative of the content. For example "Management of delayed cerebral ischaemia after aneurysmal subarachnoid haemorrhage: a narrative review"

Response 1
"We revised the manuscript to improve clarity and readability, including a modification of the title to better reflect the content. "

Comment 2
"Prevention and treatment of DCI are two separate entities. I think it is hard to follow what is prevention and what is treatment when the two are presented together. I suggest having separate sections for prevention or prophylaxis, and treatment for established DCI."

Response 2
"The sections on the prevention and treatment of DCI were separated for improved organization and focus. "

Comment 3
"A figure explaining the possible etiological mechanisms underlying DCI would be very useful."

Response 3
"We created a figure summarizing the key factors involved in DCI pathogenesis."

Comment 4
"Two tables summarising the key takeaway points from the literature for both prevention and treatment of DCI would be useful."

Response 4
"We created a table compiling the findings of previous studies on DCI prevention."

Comment 5
"A paragraph or two, plus a table, outlining key research priorities for DCI in aSAH must be included."

Response 5
"A new section outlining future directions for both the prevention and treatment of DCI was also added."

Comment 6
"A section on outcomes should be considered. What are the important outcomes after aSAH? Is change in radiological features (eg improved radiological spasm) a sufficient outcome, or should we be demanding 6-month neurological and functional outcomes?"

Response 6
"While many studies have used the presence or absence of cerebral vasospasm as the primary endpoint, we argue that therapeutic efficacy should be judged based on improvements in patient outcomes. Therefore, we emphasized the need to shift the primary endpoint toward clinically meaningful outcomes, and we included this as a key point in the future directions section."

Comment 7
"The potential role of a multinational platform trial for aSAH, modelled along the lines of REMAP-CAP (www.remapcap.org) could be discussed."

Response 7
"We also discussed the REMAP-CAP platform as a promising approach; however, a significant challenge lies in the fact that drug approvals for cerebral vasospasm vary across countries. Generating high-level international evidence for therapies that improve outcomes in patients with subarachnoid hemorrhage requires careful consideration of regulatory disparities, which remains a major barrier."

Comment 8
"A concluding statement should be provided at the end."

Response 8
"We added a conclusion section to summarize the key messages and implications of the review." 

Reviewer 4 Report

Comments and Suggestions for Authors

the paper focuses on possible druggable predictors of Delayed Cerebral Ischemia, a major contributor of poor outcome after  aneurysmal subarachnoid hemorrhage. This is a narrative review, but the time interval of the reviewed works and also the research methods used (keywords, databases etc.) are missing. No methods are reported.

Similarly there is no conclusion section.

In my opinion it would have been interesting to have also a review about possible novel  biomarkers (imaging, blood, CSF biomarkers) also coming from the omics methodologies (metabolomics, proteomics, genomics, etc).

Author Response

Comment 1
"The paper focuses on possible druggable predictors of Delayed Cerebral Ischemia, a major contributor of poor outcome after  aneurysmal subarachnoid hemorrhage. This is a narrative review, but the time interval of the reviewed works and also the research methods used (keywords, databases etc.) are missing. No methods are reported."

Response 1
"The structure of the manuscript was revised in accordance with the PRISMA guidelines, and the inclusion and exclusion criteria for study selection were clearly defined."

Comment 2
"Similarly there is no conclusion section."

Response 2
"A conclusion section was added to highlight the key findings and implications of the review."

Comment 3
"In my opinion it would have been interesting to have also a review about possible novel  biomarkers (imaging, blood, CSF biomarkers) also coming from the omics methodologies (metabolomics, proteomics, genomics, etc)."

Response 3
"The suggestion regarding future research directions is highly appreciated. Although this study did not include basic science literature within its scope, we recognize its importance and plan to incorporate such studies in future work."

Round 2

Reviewer 3 Report

Comments and Suggestions for Authors

Thank you for addressing the comments raised in my review. Congratulations on producing a very useful literature summary and update on DCI.

Comments on the Quality of English Language

Please check the language thoroughly.

There are a few instances of re-expanding acronyms after they have been defined- this is unnecessary.

There are some instances where the language could be made more formal and scientific. e.g., "formidable" could be replaced with severe or challenging. 

Author Response

Comment 1
"Please check the language thoroughly.
There are a few instances of re-expanding acronyms after they have been defined- this is unnecessary.
There are some instances where the language could be made more formal and scientific. e.g., "formidable" could be replaced with severe or challenging."

Response 1
"Abbreviations have been expanded and revised to be more academic."

Reviewer 4 Report

Comments and Suggestions for Authors

The manuscript has been heavily modified according to reviewers' comments. The structure, aims, methods of searching and selecting the literature, as well as the conclusions are now in line with the standards of scientific journals.

There is still a point that needs to be addressed:

Page 4 line 105: a reference is missing (e.g., Wolf et al, 2023)

Author Response

Comment 1
"The manuscript has been heavily modified according to reviewers' comments. The structure, aims, methods of searching and selecting the literature, as well as the conclusions are now in line with the standards of scientific journals.
There is still a point that needs to be addressed:
Page 4 line 105: a reference is missing (e.g., Wolf et al, 2023)"

Response 1
"The necessary references have been added."